



# How temperature seasonality drives interglacial permafrost dynamics: Implications for paleo reconstructions and future thaw trajectories

Jan Nitzbon[1,2,3], Moritz Langer[2,4], Luca Alexander Müller-Ißberner[3], Elisabeth Dietze[3], and Martin Werner[1]

[1]Paleoclimate Dynamics Section, Alfred Wegener Institute Helmholtz Centre for Polar and Marine Research, Bremerhaven, Germany
[2]Permafrost Research Section, Alfred Wegener Institute Helmholtz Centre for Polar and Marine Research, Potsdam, Germany
[3]Geographical Institute, Georg-August-Universiät Göttingen, Göttingen, Germany
[4]Earth sciences, Vrije Universiteit Amsterdam, Amsterdam, The Netherlands

**Correspondence:** Jan Nitzbon (jan.nitzbon@awi.de)

**Abstract.** Various proxy records have suggested widespread permafrost degradation in northern high latitudes during interglacial warm climates, including the mid Holocene (MH, 6000 years before present) and the last interglacial (LIG, 127 ka BP), and linked this to substantially warmer high-latitude climates compared to the pre-industrial period (PI). However, most Earth system models suggest only modest warming or even slight cooling in terms of annual mean surface temperatures during

these interglacials, seemingly contradicting the reconstructions of widespread permafrost degradation. Here, we combine paleo climate simulations of the Alfred Wegener Institute's Earth system model version 2.5 (AWI-ESM-2.5) with the CryoGridLite permafrost model to investigate the ground thermal regime and freeze-thaw dynamics in northern high-latitude land areas during the MH and the LIG in comparison to the PI. Specifically, we decompose how the annual mean and seasonal amplitude (that is, the difference between the maximum and minimum monthly mean) of surface temperatures affect the occurrence of

permafrost, seasonal frost, thaw depths and durations, and thermal contraction cracking activity. Our simulations reveal that (i) local permafrost probabilities and global permafrost extent are predominantly determined by mean surface temperatures, (ii) maximum thaw depths are increasing with both annual mean and seasonal amplitudes, and (iii) thermal contraction cracking within the permafrost domain is almost solely driven by the seasonal amplitude of surface temperatures. Thus, not only mean warming, but also the enhanced seasonal temperature amplitude due to a different orbital forcing have driven permafrost and

ground ice dynamics during past interglacial climates. Our results provide an additional explanation of reconstructed periods of marked permafrost degradation in the past, which was driven by deep surficial thaw during summer, while colder winters allowed for permafrost persistence in greater depths. Our results further suggest that past interglacial climates have limited suitability as analogues for future permafrost thaw trajectories, as rising mean temperatures paralleled by decreasing seasonal amplitudes expose the northern permafrost region to magnitudes of thaw that are likely unprecedented since at least Marine

Isotope Stage 11c (about 400 ka BP).



# 1   Introduction

During the Quaternary period, Earth's climate underwent substantial climatic fluctuations, marked by alternations of glacials and interglacials over the past several hundred thousand years (Köhler and van de Wal, 2020). These cycles have profoundly influenced not only the abundance of ice sheets and glaciers, but also the distribution and extent of permafrost in northern mid-

and high-latitude land areas (Willeit and Ganopolski, 2015; Lindgren et al., 2018; Saito et al., 2022). During colder climates, organic carbon became freeze-locked within the permafrost deposits, which is hypothesized to have contributed to deglacial warming through the release of greenhouse gases as these permafrost areas thaw (Tesi et al., 2016; Crichton et al., 2016; Jones et al., 2023). Conversely, the retreat of the ice sheets during deglaciations has also facilitated the emergence of new carbon sinks (Lindgren et al., 2018), highlighting the complex interactions between the climate, the cryosphere, terrestrial landscape

dynamics, and the carbon cycle (Treat et al., 2019; Brosius et al., 2021; Jones et al., 2023).

Earlier direct and indirect reconstructions of northern hemisphere permafrost extent have suggested markedly smaller permafrost areas during the MH and the LIG compared to the PI (Zoltai, 1995; Anisimov and Nelson, 1996; van Vliet-Lanoë and Lisitsyna, 2001), linking warmer climatic conditions during these periods with substantial permafrost retreat. However, the magnitude, patterns, and dynamics of climatic changes during these interglacial periods remain uncertain to date. On the

one hand, proxy records may be biased towards seasons and do not represent mean warming or cooling (Marcott et al., 2013; Bova et al., 2021; Osman et al., 2021; Erb et al., 2022). On the other hand, there are also systematic deviations between model and proxy data, as well as substantial inter-model spreads Brierley et al. (2020); Otto-Bliesner et al. (2021). For instance, there is no consensus on the magnitude and evolution of global and high-latitude temperatures during the Holocene, with proxy reconstructions pointing to a global mid Holocene thermal optimum warmer than the present, while modelling works suggest

a steady global temperature increase throughout the Holocene with only regionally warmer but globally colder mean surface temperatures compared to the PI (Kaufman and Broadman, 2023). This "Holocene temperature conundrum" has been partly attributed to seasonal biases in the proxy records (Bova et al., 2021), and poor spatial averaging of unevenly distributed proxies (Osman et al., 2021). Despite recent progress towards resolving the conundrum through data assimilation approaches (Osman et al., 2021; Erb et al., 2022), the patterns and evolution of Holocene temperatures remain uncertain (Kaufman and Broadman,

2023). Similarly, LIG temperature reconstructions are uncertain with the additional complication that many terrestrial climate proxies in high latitudes have been vanished during the glacial period. However, there is consensus that, as a general feature, both the LIG and MH surface temperatures featured larger seasonal amplitudes in the land areas of the northern hemisphere compared to the PI (O'ishi et al., 2021; Shi et al., 2022; Brierley et al., 2020), mainly as a consequence of the greater obliquity of the Earth's orbit (i.e. the tilt of its rotational axis) (Kwiecien et al., 2022).

While much effort in paleoclimate modelling is dedicated to surface temperature reconstructions, only few works specifically addressed model-based reconstructions of permafrost and seasonal frost conditions during past climates (Liu and Jiang, 2016; Saito et al., 2022; Guo et al., 2023). Standard freezing/thawing degree-day models used for approximating permafrost conditions (Riseborough et al., 2008) already suggest that the presence and characteristics of permafrost and seasonally frozen ground are not only controlled by the mean annual temperature at the surface, but also by the seasonal temperature amplitude.





For similar annual mean temperatures, marked differences in seasonal amplitude can result in distinct permafrost characteristics including the thickness of the active layer (Riseborough et al., 2008), the duration of thaw, or the occurrence of thermal contraction cracking (Lachenbruch, 1962; Matsuoka et al., 2018). These factors in turn exert key control on hydrological, geomorphological, and biogeochemical processes (Liljedahl et al., 2016; Walter Anthony et al., 2024; Lyu et al., 2024, e.g.). Thus, the seasonal temperature amplitude emerges as a potentially critical factor when reconstructing the interplay between permafrost and environments, ecosystems, and landscape dynamics during past interglacials.

Here, we assess the effects of the mean and seasonality of surface temperatures on the ground thermal regime in northern mid- and high-latitude land areas during the last and current interglacial. Specifically, we address the following objectives:

1. To evaluate the simulated interglacial permafrost extents against proxy-based permafrost reconstructions based on paleo records of speleothem growth and pollen.

2. To quantify ground thermal characteristics including (i) the probability and extent of permafrost, (ii) the maximum annual thaw depths (active-layer thicknesses), and (iii) the thermal contraction cracking activity, in the northern hemisphere (north of 30°N) for different interglacial climates (PI, MH, and LIG).

3. To decompose the contributions of the annual mean and the seasonal amplitude of surface temperatures on these ground thermal characteristics, at the global scale as well as at continental scales (that is, North America, northern Eurasia, and southern Eurasia including the Tibetan plateau).

For this, we diagnose parameter-ensemble simulations of the dedicated permafrost model CryoGridLite (Langer et al., 2024) which are driven by climatic forcing time series of different means and seasonal amplitudes of surface temperatures, generated from the output of the fully coupled Earth system model (ESM) AWI-ESM-2.5 (Shi et al., 2022). To provide a perspective on potential future thaw trajectories, we contextualize our findings for the interglacial climates with different future climate scenarios.

## 2 Methods

### 2.1 Simulations of interglacial climates using AWI-ESM-2.5

We used the coupled Earth system model AWI-ESM-2.5 (Sidorenko et al., 2019; Shi et al., 2020, 2022) at an atmosphere resolution of T63 (192 longitudes and 92 latitudes with a resolution of $1.875° \times 1.875°$ near the equator) to simulate equilibrium climates under the boundary conditions of the Paleoclimate Model Intercomparison Project Phase 4 (PMIP4) for the PI, MH, and LIG (Otto-Bliesner et al., 2021) (Table 1). The experiments were run with dynamic vegetation, and otherwise identical input datasets (e.g., glacier mask, topography, etc.) reflective of PI conditions. The PI simulation was run for a total of 3000 years. After 1600 years, the MH and LIG simulations were branched off and integrated for another 1400 years. The establishment of a quasi-equilibrium state was confirmed in all experiments by a diminishing trend in global sea surface temperatures of less than 0.05 K per century (Zhang et al., 2013).





**Table 1.** Greenhouse gas concentrations and orbital parameters used as boundary conditions for the AWI-ESM-2.5 experiments, following the PMIP4 protocol (Otto-Bliesner et al., 2017). Global-mean surface temperature ($\bar{T}_a^{\,global}$) and global-mean surface temperature seasonality ($\tilde{T}_a^{\,global}$) are shown as well.

| Experiment | Greenhouse gas concentrations | | | Orbital parameters | | | Climate characteristics | |
|---|---|---|---|---|---|---|---|---|
| | $CO_2$ (ppm) | $CH_4$ (ppb) | $N_2O$ (ppb) | Eccentricity | Obliquity (°) | Perihelion (°) | $\bar{T}_a^{\,global}$ (°C) | $\tilde{T}_a^{\,global}$ (K) |
| PI | 284.3 | 808.2 | 273.0 | 0.016764 | 23.4598 | 100.338 | 14.21 | 11.4 |
| MH | 264.4 | 597.0 | 262.0 | 0.018682 | 24.1058 | 0.878 | 13.88 | 12.1 |
| LIG | 275.0 | 685.0 | 255.0 | 0.039378 | 24.0408 | 275.418 | 14.06 | 12.7 |

From each experiment, we used the last 100 simulation years to construct a time series of daily mean surface temperatures (variable `tsurf` in ECHAM6, hereafter referred to as $T_a$) and daily snowfall rates (variable `prsn` in ECHAM6) for all 2403 land grid cells northward of 30°N to be used as climatic forcing for the CryoGridLite permafrost model (see Sect. 2.2). Figure 1 provides an overview of the mean and seasonal amplitude of surface temperatures in the simulation domain. The seasonal
90  amplitude was taken as the difference between the highest and lowest monthly mean surface temperature.

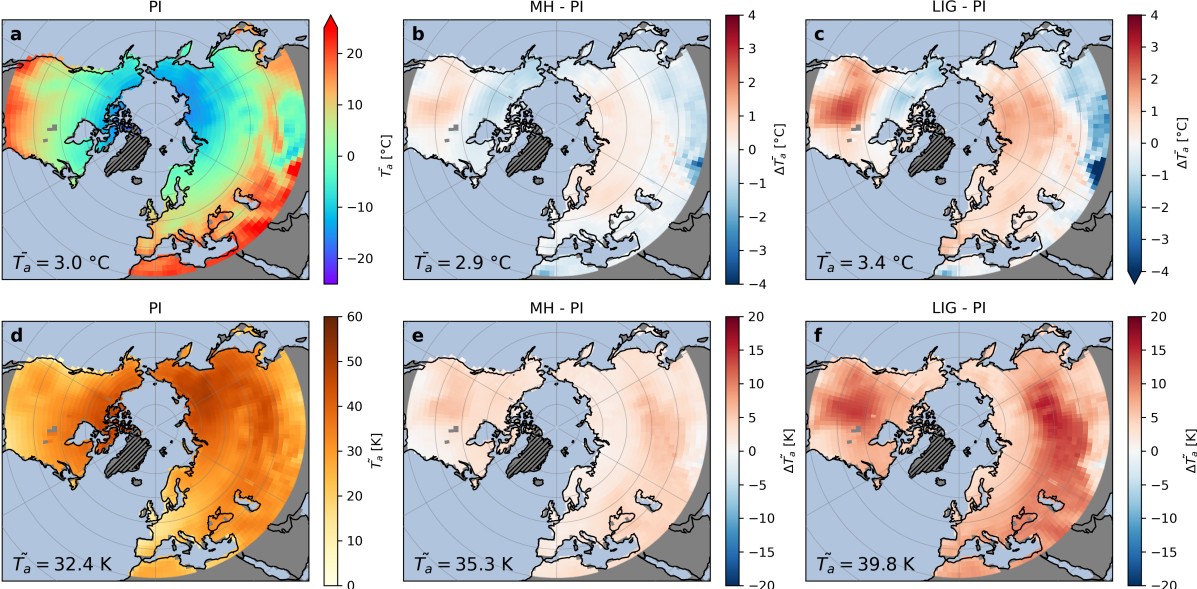

**Figure 1.** Overview of the mean ($\bar{T}_a$) and seasonal amplitude ($\tilde{T}_a$) of surface temperatures in the simulation domain. Absolute values are shown for the PI (a,d), and anomalies relative to the PI are shown for the MH (b,e) and the LIG (c,f).

In addition, we created six alternative climate forcing time series in which we combined the mean temperature and the seasonal temperature amplitude of the original PI, MH, and LIG experiments. For each original experiment (PI, MH, and LIG) we applied the temperature anomalies with respect to each of the other two experiments to the entire time series of daily mean





surface temperatures ($T_\mathrm{a}(x,y,t)$), that is, for each grid point $(x,y)$:

$$T_a^{\bar{\alpha}\tilde{\beta}}(x,y,t) = T_a^{\bar{\beta}\tilde{\beta}}(x,y,t) + \left(\left\langle T_a^{\bar{\alpha}\tilde{\alpha}}\right\rangle_t(x,y) - \left\langle T_a^{\bar{\beta}\tilde{\beta}}\right\rangle_t(x,y)\right) \tag{1}$$

where $\alpha,\beta \in \{\mathrm{PI,MH,LIG}\}$, $\bar{\cdot}$ denotes the annual mean, $\tilde{\cdot}$ the seasonal amplitude, and $\langle\cdot\rangle_t$ denotes the temporal mean over the entire forcing time series. In doing so, the seasonal amplitudes of the original forcing were retained, while the mean temperatures were adjusted to those of the other climate states. The snowfall time series were left unchanged and therefore correspond to those of the climate characterizing the seasonal amplitude. In total, we obtained nine climate forcing time series corresponding to all combinations of the means and seasonal amplitudes of the PI, MH, and LIG surface temperatures.

## 2.2 Ensemble-simulations of ground thermal regimes using CryoGridLite

### 2.2.1 Model description and input data

We used the climate forcing time series generated from the AWI-ESM-2.5 output to drive dedicated simulations with the CryoGridLite permafrost model (Nitzbon et al., 2023; Langer et al., 2024). CryoGridLite simulates subsurface heat conduction under consideration of the phase change of water using daily-mean surface temperatures as an upper boundary condition. It further simulates the dynamic build-up, compaction, and ablation of a snowpack using daily snowfall rates as input. While the version of CryoGridLite used for this study does not simulate the surface and subsurface hydrology dynamically, it addresses the spatial variability of subsurface water/ice contents through parameter-ensemble simulations.

We created an ensemble of size $n_\mathrm{ens} = 100$ of parameters and ground stratigraphies reflecting subgrid-scale variability of surface and subsurface environmental conditions. Following Nitzbon et al. (2023) and Langer et al. (2024), the ground stratigraphy was varied within the ensemble by uniformly sampling the water/ice contents in (i) the root zone between the wilting point and the field capacity ($\theta_\mathrm{R} \in [\theta_\mathrm{wp}, \theta_\mathrm{fc}]$), (ii) the vadose zone between the field capacity and the porosity ($\theta_\mathrm{V} \in [\theta_\mathrm{fc}, \phi]$), and (iii) the bedrock zone, where the water/ice content was reduced to a value $\theta_\mathrm{B} \in [0, \phi]$ while increasing the mineral content to retain saturated conditions. To represent the spatial variability of ground surface temperatures, we randomly drew from a uniform distribution (i) a maximum snow height $h_\mathrm{snow,max} \in [0.1, 2.0]\,\mathrm{m}$), (ii) a snowfall multiplier $f_\mathrm{sn} \in [0.5, 1.5]$, and (iii) a thawing-season temperature scaling factor $n_\mathrm{t} \in [0.5, 1.0]$ (Kropp et al., 2020).

Compared to the CryoGridLite simulations in Nitzbon et al. (2023) and Langer et al. (2024), we extended the simulation domain from the contemporary permafrost region to the land area northward of $30°\mathrm{N}$. For the contemporary permafrost domain, we obtained soil organic carbon contents from the dataset by Hugelius et al. (2014), which represents the upper $3.0\,\mathrm{m}$ of the subsurface. Outside the contemporary permafrost region, we used the dataset by FAO and ITPS (2018) which only represents carbon stocks in the upper $0.3\,\mathrm{m}$ of the subsurface. Otherwise, we used the same input datasets as described in Langer et al. (2024) which have global coverage. All input datasets were regridded to a Cartesian grid with a spatial resolution of $1.875° \times 1.875°$.





### 2.2.2 Experiments

We performed nine model experiments which used identical ensembles of ground stratigraphies and parameters, but different forcing time series representing all combinations of means and seasonal amplitudes of surface temperatures in the PI, MH, and LIG as described above (Fig. 2). The ground temperatures of each simulation were initialized based on an equilibrium temperature profile. First, we ran a 200-year simulation for the PI, MH, and LIG by repeating each of the original forcing time series twice. Each of these simulations was then continued for another 200 years by repeating the original forcing time series

twice. In addition, we branched off two further 200-year simulations, in which we repeated the alternative forcing time series of the same mean but different amplitudes twice. The first 300 years of each experiment were considered as spin-up to obtain a dynamic equilibrium of the ground thermal regime within at least the upper $\approx 10\,\mathrm{m}$ of the subsurface. The final 100 years of each experiment were used for further analysis.

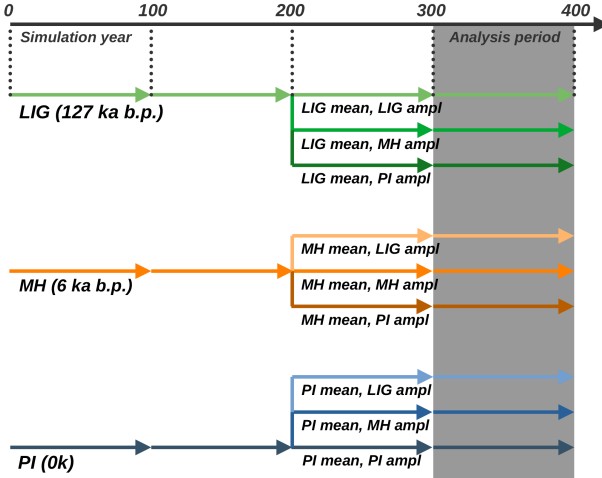

**Figure 2.** Overview of the simulation protocol. After a 200-year spin-up of each original climate forcing (PI, MH, LIG), three experiments, in which the climate forcing had the same mean temperatures, but different seasonal amplitudes were branched off and continued for another 200 years. The final 100-year period was the analysis period.

### 2.2.3 Diagnostics

We diagnosed different characteristics of the ground thermal regime which are of relevance to various processes and feedbacks acting in periglacial landscapes. First, we diagnosed the occurrence of permafrost as follows:

- We determined the *permafrost probability* ($p_{\mathrm{PF}}$) as the fraction of ensemble members for which the maximum annual temperature in $10\,\mathrm{m}$ depths (averaged over the 100-year analysis period) is below $0\,°\mathrm{C}$. Assuming the ensemble is representative of the actual spatial variability of environmental conditions, the permafrost probability corresponds



to the areal fraction of the landscape underlain by permafrost, and the overall permafrost area ($\Omega_{\mathrm{PF}}$) was obtained as $\Omega_{\mathrm{PF}} = \sum_{x,y} p_{\mathrm{PF}}(x,y)\omega(x,y)$ where $\omega(x,y)$ is the land area of grid cell $(x,y)$. The permafrost probability is a general indicator for the absence or presence of permafrost with implications for all landscape and ecosystem functions affected by permafrost, including hydrology, biogeochemistry, and geomorphology.

Based on the permafrost probabilities, we defined the joint permafrost domain as all grid cells for which $p_{\mathrm{PF}} > 0$ in any of the
nine simulations. For the joint permafrost domain, we further diagnosed the following quantities:

– The *maximum annual thaw depth* ($d_{\mathrm{thaw}}^{\max}$ (m)) was determined as the annual maximum of the depth-integrated thawed ground fraction within the upper 3 m of the subsurface (averaged over the 100-year analysis period). Thus, it is limited to a maximum of 3 m. For thaw-depths beyond 3 m, the ground can be considered to be permafrost-free. If not stated otherwise, the ensemble-mean of the maximum annual thaw depth is provided. The maximum annual thaw depth corresponds
to the depth of the active layer in permafrost areas, and has important implications for hydrology and biogeochemistry. Moreover, deeper thaw makes the initiation of thermokarst and other rapid thaw processes in ice-rich terrain more likely and can thus trigger substantial geomorphological changes.

– Similar to Stadelmaier et al. (2021), we adopted the thresholds for thermal contraction cracking provided in Matsuoka et al. (2018) to diagnose the *(deep) thermal contraction cracking activity* ($a_{\mathrm{FC}}^{\mathrm{deep}}$ (days year$^{-1}$)) as the sum of days in a
year for which the ground temperature at the surface is $T_{1\mathrm{m}} < -10\,^{\circ}\mathrm{C}$, and the temperature gradient in the upper meter of the subsurface is $\nabla T_{1\mathrm{m}} > 10\,^{\circ}\mathrm{C\,m^{-1}}$, implying a ground surface temperature of $T_{0\mathrm{m}} < T_{1\mathrm{m}} - 10\,^{\circ}\mathrm{C}$. We always report the mean over the analysis period and the ensemble. The thermal contraction cracking activity is a first order estimate for the possibility of wedge-ice formation. In these areas, excess ice growth is more likely, which can have a stabilizing effect on shorter timescales (Kanevskiy et al., 2017), but might make the landscape vulnerable to thermokarst on longer
timescales.

We further diagnosed the seasonal frost probability ($p_{\mathrm{SF}}$), the annual thawed ground fraction within the upper 3 m of the subsurface ($D_{\mathrm{thaw}}^{3\,\mathrm{m}}$), and the shallow thermal contraction cracking activity ($a_{\mathrm{FC,shallow}}$) within the joint seasonal frost domain as described in Appendix A.

## 2.3 Model evaluation against paleo records

To assess whether our model setup is capable of reproducing past interglacial permafrost extents realistically, we compared the simulated permafrost probabilities of the MH and LIG experiments to different paleo records.

### 2.3.1 Speleothem growth

First, we considered speleothem growth records, which are suitable for comparisons on glacial-interglacial time scales due to their high resistance to surface erosion (Kwiecien et al., 2022). Growth periods in the speleothem layering can be used
as an indicator for absence of permafrost, while hiatuses can suggest permafrost presence, as it precludes deep ground water





infiltration and thus inhibits speleothem growth. However, other factors can cause hiatuses or markedly modify speleothem growth as well, such as arid climates (Geyh and Heine, 2014; Nicholson et al., 2020), flooding of the cave (Denniston and Luetscher, 2017; González-Lemos et al., 2015), or glaciers (Spötl et al., 2024). For this study, we compiled speleothem growth data from different sources (Müller-Ißberner et al., 2024), including mainly the SISALv3 database (Kaushal et al., 2024) and

additional data points from individual studies (Biller-Celander et al., 2021; Batchelor et al., 2019; Vaks et al., 2013; Lauritzen, 1995; Lauritzen and Lundberg, 1999; Dublyansky et al., 2018; Žák et al., 2012; Batchelor et al., 2024). For the MH (LIG), we used all available data northward of 20°N with growth periods and hiatuses with durations longer than 500 years dated between 5 (126) ka BP and 6 (127) ka BP, obtaining a total of $n = 93$ ($n = 47$) individual locations with one or more speleothem records. We considered locations with speleothem hiatuses to be in agreement with the model if the simulated permafrost probability

was $p_{PF} > 10\%$, and locations with speleothem growth to be in agreement with the model if $p_{PF} < 90\%$, so that for the regions where the simulated permafrost coverage was sporadic or discontinuous ($10\% < p_{PF} < 90\%$), both speleothem growth and hiatuses were considered plausible.

### 2.3.2 Pollen

We additionally considered the Holocene permafrost reconstruction by Li et al. (2022), who compiled a multitude of modern
and fossil pollen records from Eurasia to establish a logistic regression model predicting permafrost coverage (classified as "permafrost-free", "non-continuous permafrost", or "continuous permafrost") based on the abundance of different pollen. Simulated permafrost probabilities of $p_{PF} \leq 10\%$ were considered to agree with the reconstruction if the pollen record indicated "permafrost-free" conditions. Probabilities of $p_{PF} \geq 90\%$ were associated with "continuous permafrost", and $10\% < p_{PF} < 90\%$ with "non-continuous" permafrost.

## 3  Results

### 3.1  Comparison between simulated and proxy-based interglacial permafrost extents

#### 3.1.1  Speleothem growth

For both the MH and LIG, there is only one location out of 83 (33), located on the southern Tibetan plateau, which is showing speleothem growth during the MH (Cai et al., 2012) and LIG (Cai et al., 2010), while our simulations suggests a high permafrost
probability $> 90\%$ (Fig. 3 a,d). This can be explained by the location of the cave in a valley surrounded by high mountain ranges. The extreme spatial variability of surface temperatures associated with the topography is not captured by the AWI-ESM-2.5/CryoGridLite setup. In addition, one location, located in the Nahanni plateau region of northwest Canada (Biller-Celander et al., 2021; Batchelor et al., 2024), shows both speleothems with growth and with hiatuses during the LIG, while our simulations suggest a high permafrost probability (Fig. 3 e). All other records of speleothem growth are plausibly located
outside the region where the simulated permafrost probabilities are $> 90\%$.





**Figure 3.** Simulated permafrost probabilities for the MH (a,b,c) and LIG (d,e,f) together with permafrost proxies based on speleothem growth (diamond marker) and pollen records (circles). Speleothem growth indicates permafrost absence while hiatuses can be explained by permafrost presence. The mixed category represents locations with records of both speleothem growth and hiatuses during the respective period. The dotted line at 30°N represents the southern limit of the simulation domain. Note that the pollen records only cover northern Eurasia during the MH.





Vice versa, almost all speleothem hiatuses in the latitudes northward of 50°N are consistently located within the region where the simulated permafrost probabilities are $> 10\%$. One exception is a record in northern Scandinavia during the LIG, which is close to the simulated permafrost region. This deviation could be explained by subglacial flooding of the cave galleries during the penultimate deglaciation (Lauritzen, 1995). Several hiatuses in lower latitudes (southward of 50°N) are located far outside

the simulated permafrost region. These can likely be attributed to either other origins like aridity, flooding, ice sheet remnants, or mountain permafrost settings that are not resolved in our model. Overall, the southern limit of the Arctic permafrost region simulated with AWI-ESM-2.5/CryoGridLite is broadly consistent with the speleothem records. However, due to its relatively sparse spatial coverage, the speleothem record does not provide a tight constraint on the past interglacial permafrost extents.

### 3.1.2 Pollen

We used the pollen-based reconstruction of Li et al. (2022) for northern Euraisa during the MH to partly compensate for the sparse spatial coverage of speleothem records. For the "continuous permafrost" category, $58$ out of $72$ records are located within the region where simulated permafrost probabilities are $> 90\%$ (Fig. 3 a,c). Most of the $14$ misaligned records are located in eastern and southern Siberia, mostly within regions with simulated non-continuous permafrost. For the "permafrost-free" category, $20$ out of $46$ records are located within the regions where permafrost probabilities are $< 10\%$, with most of

the unaligned records located within the transitional area where simulated permafrost is non-continuous. This could indicate a slight southward overestimation of the simulated Eurasian permafrost extent. For the "non-continuous" permafrost category the agreement is the lowest, with only $10$ out of $33$ records within areas with a permafrost probability between $10\%$ and $90\%$. However, most of the non-continuous permafrost records are clustered in the central Asian region between Kazakhstan, Russia, China and Mongolia, for which the model mostly simulates permafrost-free conditions. This could partly be related

to high-altitude settings not resolved in the model. As there are few pollen records suggesting non-continuous permafrost in western and eastern Siberia, the pollen-based regression model for "non-continuous" permafrost does not provide a conclusive constraint on the southern permafrost limit. Overall, the pollen-based permafrost reconstruction agrees relatively well with the simulated permafrost probabilities, even though the "permafrost-free" records might suggest a slight southward overestimation of the Eurasian permafrost extent in our model.

## 3.2 Simulated interglaical ground thermal characteristics

### 3.2.1 Permafrost and seasonal frost probability and extent

For the PI, we simulated a northern hemisphere permafrost area of $18.8 \times 10^6 \, \text{km}^2$ (Fig. 4 a). For the MH, the simulated permafrost area ($19.4 \times 10^6 \, \text{km}^2$) is higher than for the PI, with the permafrost area extending further southward in its central part of North America and in eastern Siberia, as well as in parts of the Tibetan plateau, while it is slightly smaller in western

Siberia and northern Fennoscandia (Fig. 4 e). For the LIG, the simulated permafrost area of $17.8 \times 10^6 \, \text{km}^2$ is smaller than those simulated for the PI and MH, mainly due to a reduced extent in western and central Siberia and northern Fennoscandia, while the southernmost parts of Eurasia show higher permafrost probabilities with similar patterns as for the MH (Fig. 4 i).





On the North American continent, the difference in permafrost areas between the LIG and the PI is modest, with some regions showing more, and some less permafrost.

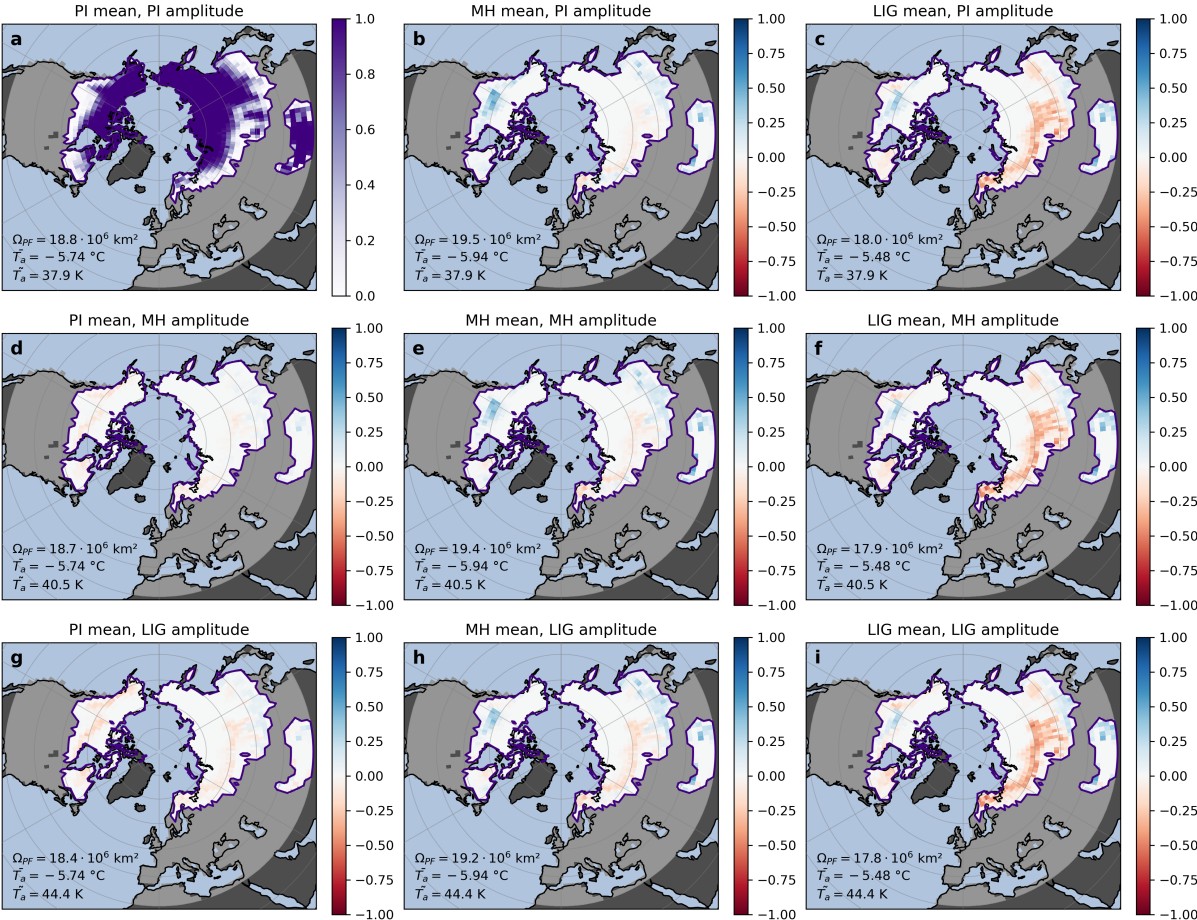

**Figure 4.** Simulated permafrost probabilities ($p_{\mathrm{PF}}$) for all climate forcings, shown as absolute values for the PI experiment in panel a, and as anomalies for all other experiments. The results for the original climate forcings are shown on the on the diagonal (PI: a; MH: e; LIG: i), and the off-diagonal panels show results for the alternative climate forcings, with each column having the same mean temperature (PI: a,d,g; MH: b,e,h; LIG: c,f,i) and each row having the same seasonal temperature amplitudes (PI: a,b,c; MH: d,e,f; LIG: g,h,i). The purple line marks the limits of the joint permafrost domain of all simulations. The permafrost area ($\Omega_{\mathrm{pf}}$), the mean annual temperature ($\bar{T}_{\mathrm{a}}$), and the mean seasonal temperature amplitude ($\tilde{T}_{\mathrm{a}}$) for the joint permafrost domain are given in the lower left corner of each panel.

Considering the alternative climate forcings, we found almost identical patterns of permafrost probabilities for those climate forcings which have the same mean temperatures, that is, for each of the columns in Fig. 4. At the same time the patterns of permafrost probabilities for identical seasonal amplitudes differ in the same way as those of the original (unbiased) forcings (each row in Fig. 4). These patterns suggest, that the (local) permafrost probability is primarily affected by the mean surface temperature, while the seasonal amplitude has only a modest influence.



The primary influence of mean temperature on permafrost presence was also indicated by high negative correlations between the mean temperatures and permafrost area at the global scale and at continental scales (Fig. 5 e-h). Across all nine simulations, the permafrost extent and mean temperature were strongly correlated at the hemispheric scale ($r = -0.97$), as well as for northern Eurasia ($r = -0.97$), the Tibetan plateau ($r = -0.95$), and the North American permafrost region ($r = -0.87$). The respective correlations between permafrost extent and seasonal temperature amplitude are all negative, but fairly low (northern hemisphere: $r = -0.21$; northern Eurasia $r = -0.06$; Tibetan plateau: $r = -0.05$), with the exception of North America ($r = -0.46$, Fig. 5 h), where the simulated permafrost extent tended to be lower in climates with a higher seasonal amplitude.

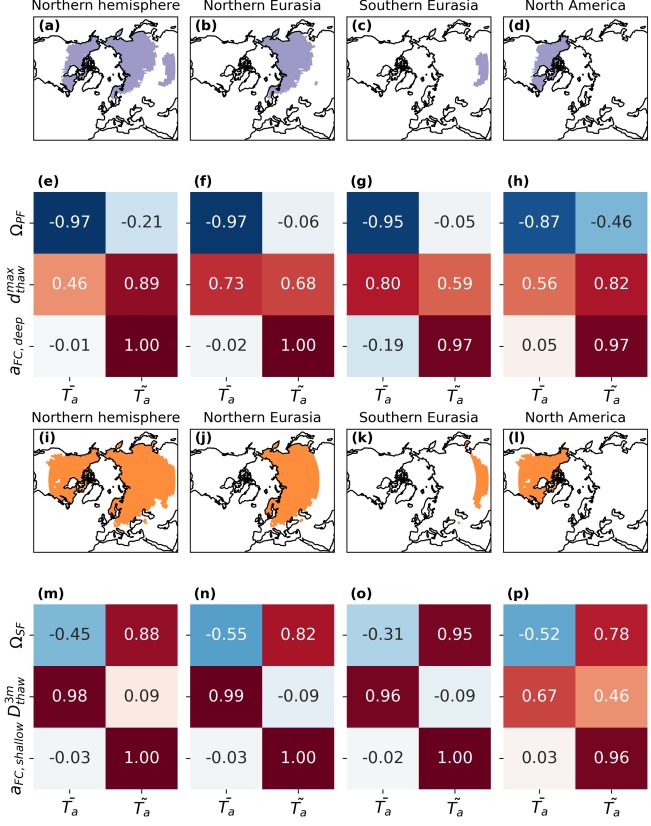

**Figure 5.** Northern hemisphere and regional Pearson correlations between different diagnostics and the mean temperatures ($\bar{T}_a$) and the seasonal temperature amplitudes ($\tilde{T}_a$) of all nine experiments. Panels a-d and i-l respectively show the part of the joint permafrost and seasonal frost domains which are considered for the correlation analysis in the panels below (northern hemisphere: lat>30°N; northern Eurasia: lat>40°N, 0°E<lon<186°E; southern Eurasia: lat<40°N, 0°E<lon<186°E; North America: lat>30°N, 186°E<lon<360°E). For the joint permafrost domain (e-h), we considered the permafrost area ($\Omega_{PF}$), the maximum thaw depth ($d_{thaw}^{max}$), and the deep thermal contraction cracking activity ($a_{FC,deep}$). For the joint seasonal frost domain (m-p), we considered the seasonal frost area ($\Omega_{SF}$), the thawed ground fraction ($D_{thaw}^{3m}$), and the shallow thermal contraction cracking activity ($a_{FC,shallow}$).





For seasonal frost area (including permafrost), we also found a negative correlation with mean temperatures (northern hemisphere: $r = -0.45$), but also a marked positive correlation with seasonal amplitude ($r = 0.88$) (Fig. 5 m), and comparable correlation magnitudes regionally (Fig. 5 n-p). As visible from Fig. A1, higher seasonal amplitudes of the MH and LIG increase the seasonal frost probability outside the permafrost domain, while the higher mean temperatures of the LIG cause a reduction of $p_{SF}$. These opposing effects are partly compensating each other in the original MH and LIG climates (Fig. A1 e,i).

### 3.2.2  Maximum thaw depth and thawed ground fraction

The results for the maximum annual thaw depth (i.e., the active layer thickness in permafrost areas) reveal more complex patterns and dependencies on mean temperatures and seasonal temperature amplitudes than those for the permafrost probabilities (Fig. 6). The average maximum thaw depth across the joint permafrost region was $1.61\,\mathrm{m}$ in the PI and thus lower than in the MH ($1.64\,\mathrm{m}$) and LIG ($1.80\,\mathrm{m}$).

For the MH, maximum thaw depths are larger (deeper active layers) compared to the PI across northern Eurasia and particularly in northern Fennoscandia and western Siberia, while they are smaller (shallower active layers) in more southern latitudes including south-eastern Siberia and the Tibetan plateau (Fig. 6 e). In North America, the MH-PI thaw-depth anomalies are modest. For the LIG, we simulated substantially larger thaw depths across the high-latitude permafrost region, with marked increases along its southern margin in Eurasia, coinciding with the regions where permafrost extent is smaller (Fig. 4 i). Only for parts of the Tibetan plateau and south-eastern Siberia, the thaw depths are shallower in the LIG climate compared to the PI reference.

For the alternative climate forcings, we found different regional patterns. On the North American continent, thaw depths are substantially increasing for the higher seasonal amplitudes of the MH and LIG climates when the mean temperatures are fixed at PI levels (Fig. 6 d,g). When the PI seasonal temperature amplitude was fixed and the mean temperatures shifted, thaw depths were substantially smaller for both the MH and LIG mean temperatures (Fig. 6 b,c), consistent with the PI climate being the warmest simulated for the North American permafrost region. These opposing trends for the North American continent more or less cancel each other out in the other simulations (Fig. 6 e,f,h), with only the original LIG climate showing substantially deeper thaw than the PI. In northern Eurasia, both the mean and the amplitude are increasing from PI over MH to LIG climate, resulting in substantially deeper thaw depths than for the PI in this region in all simulations. In southern Eurasia including the Tibetan plateau, deeper thaw depths are associated with the higher seasonal amplitudes of the MH and LIG climates (Fig. 6 d,g), while shallower thaw depths were simulated for the MH and LIG mean temperatures, as these are colder in this region (Fig. 6 b,c). Similar to the pattern found for North America, these effects are partly cancelling each other out in the remaining simulations (Fig. 6 e,f,h,i).

The correlation analysis for the northern hemisphere joint permafrost region (Fig. 5 a) confirmed that the maximum thaw depth is positively correlated with both the mean ($r = +0.46$) and the seasonal amplitude ($r = +0.89$) of the surface temperatures (Fig. 5 e). Similar correlation values were found for the North American permafrost region ($r = +0.56$ for the mean; $r = +0.82$ for the amplitude; Fig. 5 h). On the Eurasian continent, the maximum thaw depth showed a slightly stronger pos-



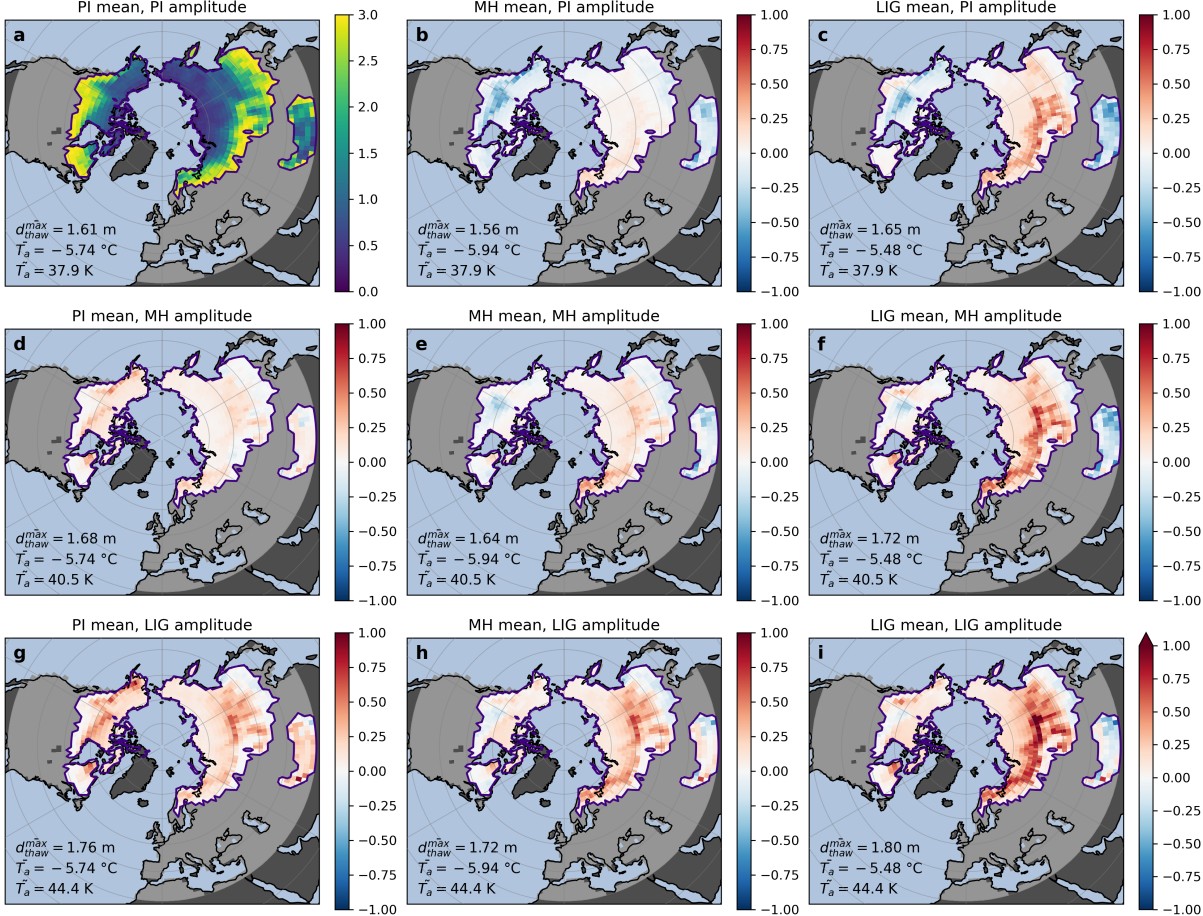

**Figure 6.** Same as Fig. 4, but for the simulated maximum thaw depths ($d_{\text{thaw}}^{\max}$). The mean maximum thaw depth ($d_{\text{thaw}}^{\bar{\max}}$), the mean annual temperature ($\bar{T}_a$), and the mean seasonal temperature amplitude ($\tilde{T}_a$) for the joint permafrost domain are given in the lower left corner of each panel.

280 itive correlation with mean temperatures (Eurasia: $r = +0.73$; Tibetan plateau: $r = +0.80$) than with temperature amplitude (Eurasia: $r = +0.68$; Tibetan plateau: $r = +0.59$; Fig. 5 f,g).

As the maximum thaw depth only captures the deepest extent of thaw but not its duration, we conducted the same analysis also for the annual thawed ground fraction ($D_{\text{thaw}}^{3m}$) as defined in Appendix A (Fig. A2). In contrast to the maximum thaw depth, the annual thawed ground fraction is primarily affected by the annual mean temperature (northern hemisphere: $r = 0.98$), while

285 the seasonal temperature amplitude has a minor effect ($r = 0.09$; Fig. 5 m). Similar correlation scores were found for northern and southern Eurasia (Fig. 5 n,o), while for North America, also the seasonal amplitude showed a clear positive correlation with thawed ground fraction ($r = 0.46$, Fig. 5 p).





### 3.2.3 Thermal contraction cracking activity

In all simulations, deep thermal contraction cracking was most probable in the continental parts of Eurasia including northern
and southern central Siberia (Fig. 7). The overall deep thermal contraction cracking activity was lowest in the PI simulation
($a_{\mathrm{FC,deep}}^- = 0.34$), substantially higher for the MH ($a_{\mathrm{FC,deep}}^- = 0.45$), and almost twice as high for the LIG ($a_{\mathrm{FC,deep}}^- = 0.61$).

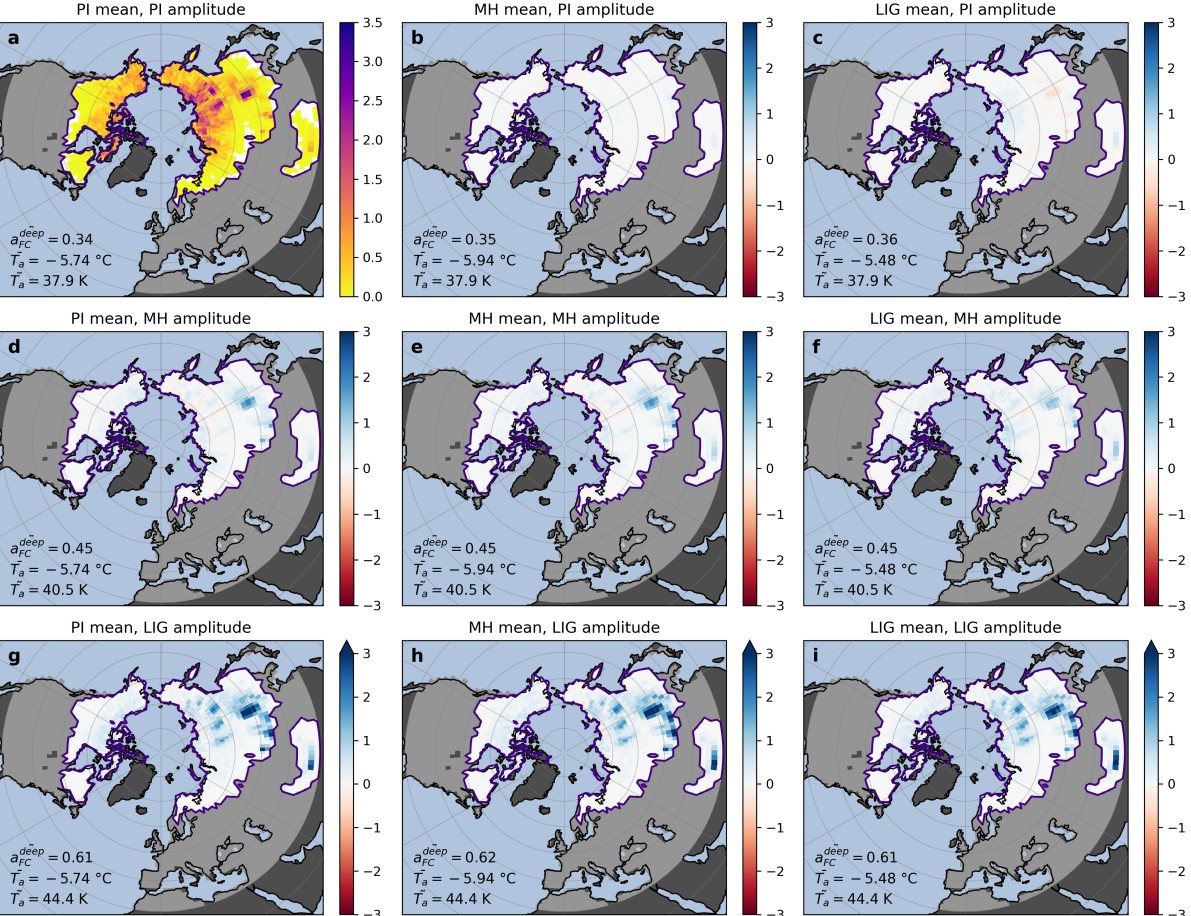

**Figure 7.** Same as Fig. 4, but for the deep thermal contraction cracking activity ($a_{\mathrm{FC}}^{\mathrm{deep}}$). The mean deep thermal contraction cracking activity
($a_{\mathrm{FC}}^{\overline{\mathrm{deep}}}$), the mean annual temperature ($\bar{T}_{\mathrm{a}}$), and the mean seasonal temperature amplitude ($\tilde{T}_{\mathrm{a}}$) for the joint permafrost domain are given in the
lower left corner of each panel.

For each original climate forcing, changes of the mean temperatures towards those of the other climates did not result in any
significant changes in deep thermal contraction cracking patterns and magnitudes (that is, there are identical patterns for each
row in Fig. 7). Vice versa, irrespective of the mean temperatures, the deep thermal contraction cracking activity increased with
the higher amplitudes of the MH and LIG climates (cf. each column in Fig. 7), overall suggesting a predominant influence of
temperature seasonality on deep thermal contraction cracking activity.



The correlation analysis confirms, that at the hemispheric scale, deep thermal contraction cracking activity is not correlated with mean temperatures ($r = -0.01$), but very strongly positively correlated with the seasonal amplitude ($r = +1.00$) (Fig. 5 e). Similar correlations were obtained for northern Eurasia (mean: $r = -0.02$, amplitude: $r = +1.00$; Fig. 5 f) and North America (mean: $r = 0.05$, amplitude: $r = +0.97$; Fig. 5 h). For southern Eurasia, the correlation with the seasonal temperature amplitude was also very strong ($r = +0.97$), but there was also a slightly negative correlation with the mean temperatures ($r = -0.19$; Fig. 5 g), indicating that deep thermal contraction cracking becomes less probable in regions of generally warm permafrost.

Complementary results for the activity of shallow thermal contraction cracking in the joint seasonal frost domain are provided in Fig. A3. Similar to $a_{\text{FC,deep}}$, $a_{\text{FC,shallow}}$ is almost exclusively driven by the seasonal amplitude (northern hemisphere: $r = 1.00$; Fig. 5 m), while the annual mean temperature has no substantial effect in the northern hemisphere seasonal frost domain ($r = -0.03$). Similar correlation scores were found regionally (Fig. 5 n-p).

## 4 Discussion

### 4.1 Model accuracy, uncertainties, and limitations

To assess the accuracy and to identify uncertainties and limitations of our model simulations, it is instructive to compare our modelling results to contemporary observations, proxy-based reconstructions and other model simulations of permafrost characteristics. For the PI, we simulated a northern hemisphere permafrost area of $18.8 \times 10^6 \, \text{km}^2$ which exceeds the estimates of Zhang et al. (2000) (12.2 to $17.0 \times 10^6 \, \text{km}^2$; late 20th century), Gruber (2012) (13 to $18 \times 10^6 \, \text{km}^2$; 1961-1990), and Obu et al. (2019) ($13.9 \times 10^6 \, \text{km}^2$; 2000-2016) for recent historical conditions. This can partly be attributed to the PI being colder than the recent historical conditions, but also points to a slight tendency of our AWI-ESM-2.5/CryoGridLite setup to overestimate permafrost probabilities.

The higher permafrost area we simulated the MH ($19.4 \times 10^6 \, \text{km}^2$) compared to the PI is in contrast to most earlier proxy-based reconstructions and modelling. Early modelling work by Anisimov and Nelson (1996) suggested substantial northward permafrost retreat during the MH (44% reduction of permafrost region). However, the authors assumed a mean global surface warming of $+1\,^\circ\text{C}$ above recent conditions for the MH climate, which is exceeding more recent estimates (reviewed by Kaufman and Broadman (2023): $+0.54\,\text{K}$ (Marcott et al., 2013); $+0.47\,\text{K}$ (Temp12k, multi-proxy; Kaufman et al. (2020)); $+0.3\,\text{K}$ (LegacyClimate; pollen only; northern hemisphere north of 30°N; Herzschuh et al. (2023)); $+0.12\,\text{K}$ (Holocene DA; data assimilation; Erb et al. (2022)); $-0.15\,^\circ\text{C}$ (LGMR; data assimilation; Osman et al. (2021)); $-0.3\,\text{K}$ (PMIP4-CMIP6 multimodel mean; Brierley et al. (2020)). The MH-PI global temperature anomaly in our AWI-ESM-2.5 experiments amounts to $-0.33\,\text{K}$ and compares well to the lower end of the more recent estimates (Tab. 1). Similarly, the earlier proxy-based reconstructions by Zoltai (1995) and van Vliet-Lanoë and Lisitsyna (2001) suggested marked MH permafrost retreat. However, these were mostly based on vegetation-derived proxies, which can be biased towards the growing season (Bova et al., 2021; Kaufman and Broadman, 2023) and therefore overestimate warming and the associated permafrost retreat due to the higher temperature seasonality during the MH. Also the pollen-based reconstructions by Li et al. (2022) suggest a more northern permafrost limit



during the MH in eastern Eurasia (Fig. 3 a), as reconstructions of permafrost-free conditions are located within our simulated discontinuous permafrost zone. However, the pollen-based reconstructions of non-continuous permafrost by Li et al. (2022), do not provide a strong constraint in Siberia, as the corresponding locations are relatively clustered across Mongolia and north-western China, potentially in mountainous settings, where our model partly simulates permafrost absence. Compared to the reconstructions by van Vliet-Lanoë and Lisitsyna (2001), the work by Li et al. (2022) suggests a less pronounced permafrost retreat during the MH, and is thus closer to our modelling results. Our results for the MH agree relatively well with the more recent modelling work by Liu and Jiang (2016) who used the output of CMIP5-PMIP3 MH experiments to diagnose seasonal frost and permafrost extents, and active layer thicknesses in comparison to the PI. While all individual models investigated by Liu and Jiang (2016) show a relative reduction of permafrost areas for the MH (ranging between $1.8$ and $19.0\,\%$) and a relative increase in seasonal frost areas, our simulations show a relative increase in permafrost area of $3.2\,\%$ and a relative increase in seasonal frost area by $3.4\,\%$. Consistent with the model-ensemble mean by Liu and Jiang (2016), we simulated deeper active layers (relative increase by $1.9\,\%$) for the MH compared to the PI. Besides slightly different ways of diagnosing the seasonal frost and permafrost compared to Liu and Jiang (2016), the main difference of increased MH permafrost extent in our simulations can be attributed to the colder MH climate relative to the PI simulated by AWI-ESM-2.5 (Table 1).

As for the MH, our simulations are in marked contrast to the much-reduced LIG permafrost extent in Eurasia suggested by earlier modelling by Anisimov and Nelson (1996). Again, the assumed mean global surface warming of $2\,°C$ for the LIG climate by Anisimov and Nelson (1996) is exceeding more recent estimates of LIG-PI global temperature anomalies (including $+0.62\,K$ (tropical sea surface temperature proxies; Bova et al. (2021)); $+0.5\,K$ (global sea surface temperature proxies; Hoffman et al. (2017)); $-0.2\,K$ (PMIP4-CMIP6 multimodel mean; Otto-Bliesner et al. (2021)). For comparison, the LIG-PI global temperature anomaly in our AWI-ESM-2.5 experiments amounts to $-0.16\,K$ (Tab. 1). For North America during the LIG, Reyes et al. (2010) suggest widespread surficial permafrost thaw also within the current continuous and discontinuous permafrost regions, while deeper ice wedges were preserved and still persist until today. Similarly, Lauriol et al. (1997) suggests permafrost persistence in the northern Yukon during the Quaternary period. These findings align well with ours, that (i) permafrost as diagnosed via the temperature in $10\,m$ depth persisted in northern North America during the LIG (Fig. 4 i), while (ii) thaw depths were substantially deeper, potentially triggering surficial thaw as manifested in features such as thaw slumps or thaw lakes. However, the deep permafrost might also have persisted during a warm LIG climate due to ecosystem factors such as a protective vegetation (forest) cover, which is only indirectly represented via $n_t$-factors in our model.

The simulated MH and LIG permafrost probabilities are broadly consistent with speleothem-based permafrost proxies compiled from different sources (Kaushal et al., 2024; Biller-Celander et al., 2021; Vaks et al., 2013, 2020) (Fig. 3). Our simulations for the MH and LIG are consistent with the North American speleothem records discussed in Biller-Celander et al. (2021), but a more recent study by Batchelor et al. (2024) reports speleothem growth for a cave on the Nahanni plateau in northwestern Canada during the LIG (Marine Isotope Stage (MIS) 5e). Due to the sparse spatial coverage, the speleothem records do not provide a tight spatial constraint, and an actual northward-shift of the permafrost zones during both the MH and LIG is well possible. Similarly, the persistence of permafrost in the Lenskaya Ledyanaya cave in Siberia since MIS-11 (Vaks et al., 2013) is consistent with our simulations, suggesting permafrost presence in that region during the current and last interglacial. Thus,





or simulations of relatively stable permafrost presence in $10\,\mathrm{m}$ depth during the MH and LIG, broadly align with the interpretations of Biller-Celander et al. (2021) and Vaks et al. (2020) that permafrost became increasingly persistent over the Pleistocene, while speleothem growth periods associated with deep permafrost thaw date back to at least MIS-11 in Northern Yukon caves and Ledyana Lenskaya cave in central Siberia, and largely to MIS-9 and MIS-11 in caves on the Nahanni plateau northwestern Canada. These changes in permafrost persistence over the Pleistocene have been linked to increasing summer sea ice extents,

hindering heat and moisture transport from the Arctic ocean towards the terrestrial Arctic (Lawrence et al., 2008; Vaks et al., 2020). Other processes on glacial–interglacial time scales such as ecological and vegetation succession (Treat et al., 2019), carbon cycle responses (Lindgren et al., 2018), changing atmospheric or oceanic circulation patterns and associated changes in precipitation and snow (Löfverström et al., 2014) also potentially affect interglacial permafrost characteristics.

Overall, exceeding most previous estimates, our simulated interglacial continuous permafrost areas are probably slightly

overestimated. At the same time, the transition from discontinuous to sporadic to isolated permafrost is relatively sharp and hence the non-continuous permafrost area tends to be underestimated in our simulations. On the model-side, this and other deviations from observations and proxy records can be related to various factors including (i) a generally cold-biased climate in northern high latitudes simulated by AWI-ESM-2.5 Brierley et al. (2020); Otto-Bliesner et al. (2021), (ii) missing or overly simplified process representations in CryoGridLite (e.g., no soil freezing characteristic, no dynamic vegetation), (iii) the

parameter-ensemble not representing the actual variability of surface temperatures and environmental conditions within a grid cell (for example in mountainous terrain), (iv) equilibrium simulations not capturing transient climate effects during the simulated interglacial periods, and (v) uncertainties related to the way permafrost is diagnosed (Steinert et al., 2023). Despite these limitations and uncertainties, our approach allows to draw conclusions about the general influences of the mean temperature versus the seasonal temperature amplitude on the ground thermal dynamics and associated processes.

## 4.2 How temperature seasonality drives interglacial permafrost dynamics

### 4.2.1 Implications for reconstructions of paleo-climates and -environments

The mean surface temperature turned out to exert the dominant control on the presence of permafrost. This is consistent with the approaches of Gruber (2012) and Chadburn et al. (2017) who derived robust, observation-based relations between the local mean annual air temperature and the probability of equilibrium permafrost occurrence. These works and our approach

have in common, that they cannot diagnose transient effects such as disequilibrium or relict permafrost (Opel et al., 2024) which can be present in great depths or locations with highly-insulating surface covers such as in larch taiga, where also long-term vegetation–fire–permafrost–climate feedbacks play a role (Dietze et al., 2020; Stuenzi et al., 2022). With respect to reconstructions of past interglacial climates and paleo environments, our results suggest a close link between the mean surface temperatures and the presence or absence of permafrost, which would generally allow to derive information about

permafrost presence from annual temperature proxies. However, our findings also underline the importance that proxies must be reflective of annual mean temperature conditions and not biased towards seasons in order to establish reliable links. Proxies that are biased towards either summer or winter season might actually reflect differences in temperature seasonality instead





of different means (Kwiecien et al., 2022). For example, reconstructions of permafrost extent based on plant macrofossils or pollen, could be biased towards the growing season (Bova et al., 2021), and therefore suggest the absence of permafrost

during past interglacials, even though colder winters could still compensate and allow for permafrost presence. This seasonal bias problem can partly explain the differences between earlier proxy-based reconstructions suggesting marked interglacial permafrost retreat Zoltai (1995), and more recent works including ours suggesting interglacial permafrost persistence at depth (Reyes et al., 2010; Biller-Celander et al., 2021).

According to our simulations, maximum annual thaw depths are positively correlated to both, the mean and the seasonality

of surface temperatures. This finding is consistent with theoretical considerations and simple models like the Stefan model for which thaw depths increase with the number of thawing degrees days which in turn increase with both higher mean temperatures and larger seasonal amplitudes (Riseborough et al., 2008). Therefore, substantially higher thaw depths compared to PI conditions can be expected under the higher seasonal amplitudes of the MH and LIG climates, even if the mean temperatures would remain at PI levels (Fig. 6 a,d,g). As warmer summers and higher thaw depths can initiate thermokarst (e.g. thaw lake

formation) and thermo-erosion (e.g. retrogressive thaw slumps) even in cold permafrost (Farquharson et al., 2019; Jones et al., 2019), a high abundance of these features, for example during the early and the mid Holocene (Brosius et al., 2021), does not necessarily require higher mean annual temperatures, but can also be caused by warm summers associated with marked seasonal temperature amplitudes. In our AWI-ESM simulations, the MH and LIG climates are even colder than the PI in northern North America and the Tibetan plateau (Fig. 1), which seemingly contradicts reconstructions of substantial near-surface thaw

compared to today during these interglacial climates (Reyes et al., 2010). However, the effect of higher seasonal temperature amplitudes on thaw depths compensates for the colder mean temperatures during the MH, and exceeds it during the LIG, providing an explanation for abundant surficial permafrost thaw features during these periods. Therefore, past periods of enhanced permafrost thaw indicated by the occurrence of thermokarst and thermo-erosion features can at least partly be attributed to the larger seasonal temperature amplitude during those times, and do not necessarily require substantial reductions in the equilib-

rium permafrost extent, since deep permafrost might have been resilient to surficial thaw (Reyes et al., 2010; Biller-Celander et al., 2021).

Besides its positive correlation with seasonal thaw depths, we found the temperature seasonality to almost exclusively determine the abundance of thermal contraction cracking during the MH and LIG, with higher seasonal amplitudes implying more frequent cracking. As thermal contraction cracking activity drives the formation and growth of ice wedges, it is also

a first-order indicator of excess ice accumulation (Bockheim and Hinkel, 2012), which can be considered as a stabilizing process for permafrost, at least on decadal time scales (Jorgenson et al., 2015; Kanevskiy et al., 2017). On centennial to millennial time scales, high excess ice contents do, however, increase the vulnerability of the landscape to the occurrence of rapid thaw processes like thermokarst (Kokelj and Jorgenson, 2013). Therefore, the occurrence of thermokarst during past interglacials could at least partly be associated with a generally more active "ground ice cycling" compared to recent condition,

especially in continental regions with a higher seasonal temperature amplitude. With respect to past climate and environment reconstructions, our diagnostic for thermal contraction cracking (Figs. 7 and A3) based on the comprehensive observations and synthesis by Matsuoka et al. (2018) can be directly linked to the occurrence of ice-wedges, ice-wedge casts, and potentially




sand wedges (Black, 1976; Wolfe et al., 2018). Below the mean annual temperatures required for permafrost occurrence, our results indicate an absence of a strong control of the mean annual temperature on deep thermal contraction cracking activity,
suggesting that observations of ice-wedge casts or sand wedges should not be used to reconstruct mean air temperatures which is in line with earlier findings (Murton and Kolstrup, 2003; Christiansen et al., 2016). The occurrence of deep thermal contraction cracking was well aligned with the simulated permafrost domain (Fig. 7), confirming that past permafrost limits and periglacial conditions can be reconstructed based on the occurrence of deep ice-wedge casts Stadelmaier et al. (2021). However, conditions favourable for shallow thermal-contraction cracking are also found outside the permafrost domain yet
within the seasonal frost domain (Fig. A3), which has to be kept in mind when interpreting ice-wedge or sand-wedge casts.

Overall, our results suggest more active excess ground ice dynamics during the MH and LIG compared to recent conditions, and potentially also during further past interglacials. This can be attributed to a increased seasonal temperature amplitudes which on the one hand leads to more abundant thermal contraction cracking and thus enhanced ground-ice accumulation in form of ice wedges during winter, and on the other hand causes deeper thaw during summer, making the initiation of rapid thaw
processes like thermokarst and thermo-erosion more likely. These effects should be considered in proxy-based reconstructions of paleo climate and environments based on ground-ice related proxies.

### 4.2.2 Implications for the permafrost response to ongoing and future climate change

Beyond the implications for reconstructions of past climates, it is instructive to contextualize our findings with respect to projections of future climate change. While the differences between the past interglacial climates of the MH and LIG compared
to the PI are primarily driven by different orbital configurations (Tab. 1), the currently ongoing and near-future climate change is happening on decadal to centennial time scales and thus solely driven by a rapidly changing greenhouse gas forcing. Accordingly, the MH and LIG climates in the AWI-ESM and other ESMs (Brierley et al., 2020; Otto-Bliesner et al., 2021) are characterized by a substantially higher temperature seasonality, but relatively modest patterns of regional cooling and warming (Fig. 1). Conversely, the near-future climates are characterized by marked temperature increases, in particular in the high
latitudes (Rantanen et al., 2022). At the same time, asymmetric seasonal warming trends due to more rapid winter warming (Bintanja and van der Linden, 2013) are likely to lead to a decrease in temperature seasonality in both the SSP1-2.6 and SSP5-8.5 scenarios, overall setting the future climate on a trajectory much different from the states experiences during past interglacials (Fig. 8).

To highlight implications of these contrasting climate trajectories, we derived an analytical estimate of the annual thawed
ground fraction ($D_{\text{thaw}}^{\text{3m}}$) based on the Stefan model (see Appendix B; Fig. 8). Relative to the PI state, both an increase in seasonality as well as an increase in mean temperatures would lead to an increase in thawed ground fraction on average. However, while the increase is relatively small for the past interglacials, the current and near-future climate trajectory (from PI over HIST to the SSPs) corresponds to the most rapid increase, as it is oriented perpendicular to the isolines of $D_{\text{thaw}}^{\text{3m}}$. Our results further indicate that with decreasing seasonality of the projected near-future climate trajectory, the thermal contraction
cracking activity is going to decrease, while it was higher during the MH and LIG interglacials. Thus, excess ice accumulation and the associated short-term stabilization of permafrost is becoming less likely in the near future.



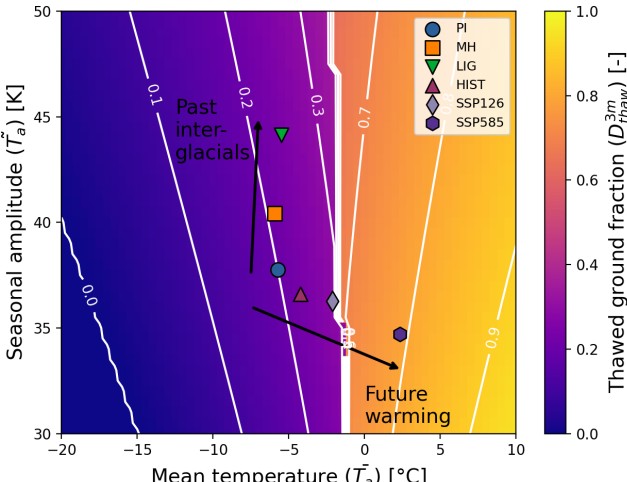

**Figure 8.** Scatter plot of the mean annual temperature ($\bar{T}_a$) and temperature seasonality ($\tilde{T}_a$) averaged over the joint permafrost domain for different climate states simulated with AWI-ESM-2 (PI: pre-industrial; MH: mid Holocene; LIG: last interglacial; HIST: historical (1990-2014); SSP126: Shared socio-economic pathway scenario 1-2.6 (2075-2099); SSP126: Shared socio-economic pathway scenario 5-8.5 (2075-2099)). The background shows the annual thawed ground fraction within the upper 3 m of the subsurface ($D_{\text{thaw}}^{3m}$) according to the Stefan model (see Appendix B for details).

Overall, our findings suggest that with respect to permafrost dynamics, the most recent interglacial climates of the MH and LIG can, if at all, only partly serve as analogues for the changes to be expected under the current and near-future climate trajectory. While the abundance of surficial permafrost thaw in those past periods can partly be attributed to the effects of
a higher temperature seasonality and thus warmer summers and deeper seasonal thaw, the current and future thaw is driven by year-round warming, which is particularly pronounced in winter (Rantanen et al., 2022). During summers, the observed circum-Arctic intensification of fire regimes could further contribute to active-layer thickening and permafrost thaw (Li et al., 2022; Jiang et al., 2023; Diaz et al., 2024), but also permafrost thaw could feed back to further intensify fires via more rapid soil drying (Kim et al., 2024). In addition, sea-ice decline on a trajectory to an ice-free Arctic during summer (Guarino et al.,
2020; Vermassen et al., 2023; Jahn et al., 2024), could drive further permafrost thaw through increased ocean-to-land heat and moisture transport (Lawrence et al., 2008; Vaks et al., 2020; Nitzbon et al., 2024). Consequently, the permafrost region is entering uncharted territories in the "climate space", which are likely associated with a decline in permafrost extent, deeper seasonal thaw and thus higher likelihood of abrupt thaw initiation, less potential for landscape stabilization through excess ice accumulation, and longer exposures of soil carbon to thawed conditions.



## 5    Conclusions

We have conducted parameter-ensemble simulations of ground thermal characteristics with the CryoGridLite permafrost model, which were driven by pre-industrial, mid Holocene, and last interglacial climate forcing time series obtained with the AWI-ESM-2.5 coupled climate model using PMIP4 boundary conditions. From the presented results we draw the following conclusions:

1. The permafrost extent simulated with AWI-ESM-2.5/CryoGridLite agrees well with speleothem- and pollen-based permafrost reconstructions for the MH and LIG, albeit a slight tendency of our model setup to overestimate permafrost extent for all climates, which is most plausibly explained by a cold-bias in the northern high-latitude surface temperatures simulated by AWI-ESM-2.5.

2. Compared to the PI, we found a modest increase in permafrost extent for the MH ($+3\,\%$), and a slightly higher decrease for the LIG ($-5\,\%$). Maximum thaw depth were found to be higher for both the MH ($+2\,\%$) and the LIG ($+12\,\%$). Thermal contraction cracking activity was found to be substantially higher during both the MH ($+32\,\%$) and the LIG ($+79\,\%$), with a clear regional signal in northern and southern Siberia.

3. There are distinct responses of permafrost to interglacial climate change between North America and Eurasia, as well as between northern and southern Eurasia. While the changes in mean temperatures and seasonal amplitudes combine to substantially higher thaw depth in northern Eurasia, they partly cancel out each other in North America and southern Eurasia.

4. Permafrost and seasonal permafrost characteristics are differently affected by mean temperatures and seasonal temperature amplitudes. Most notably, permafrost probability is predominantly controlled by the annual mean temperatures, while the seasonal amplitude almost exclusively determines the thermal contraction cracking activity within the permafrost domain.

5. Earlier reconstructions of near-surface permafrost thaw during interglacial climates can be partly explained by higher thaw depths associated with higher seasonal temperature amplitudes, and do not require higher mean annual temperatures and permafrost loss in greater depths.

As the Arctic's future climate trajectory is characterized by increasing mean temperatures and decreasing temperature seasonality, the suitability of past interglacials as analogues for future permafrost dynamics and feedbacks is very limited, and instead permafrost thaw trajectories unprecedented since at least MIS-11 (about 400 ka BP) are to be expected within the current century.

*Code and data availability.*    The original climate forcing time series based on the AWI-ESM-2.5 model output and a script to generate the alternative climate forcing time series is provided at https://zenodo.org/records/14244199 (Nitzbon, 2024a). The CryoGridLite model



code and the parameter input file used for the simulations in this study is deposited at https://zenodo.org/records/14243849 (Nitzbon and Langer, 2024). The CryoGridLite model diagnostics, a plotting script, and a script to run the semi-analytical Stefan model is provided at https://zenodo.org/records/14243857 (Nitzbon, 2024b). The speleothem growth records compiled for the model evaluation are provided at https://zenodo.org/records/14512888 (Müller-Ißberner et al., 2024).

## Appendix A: Additional diagnostics

In addition to the diagnostics described in Section 2.2.3, we complemented our analysis by the following quantities.

- We diagnosed the *seasonal frost probability* ($p_{\mathrm{SF}}$) as the fraction of ensemble members for which the minimum annual temperature in $1\,\mathrm{m}$ depths is below $0\,°\mathrm{C}$, i.e. frost regularly penetrates down to a depth of $1\,\mathrm{m}$. The overall seasonal frost area (including the permafrost area) was obtained as $\Omega_{\mathrm{SF}} = \sum_{x,y} p_{\mathrm{SF}} \omega(x,y)$.

Based on the simulated seasonal frost probabilities, we defined the joint seasonal frost domain as all grid cells for which $p_{\mathrm{SF}} > 0$
or $p_{\mathrm{PF}} > 0$ in any of the nine simulations. For the joint seasonal frost domain, we further diagnosed the following quantities, providing further insights into the ground thermal regime beyond the permafrost domain:

- Following Harp et al. (2016), we additionally diagnosed the annual thawed ground fraction ($D_{\mathrm{thaw}}^{3\mathrm{m}}$) as the depth- and time-integrated fraction of thawed ground within the upper $3\,\mathrm{m}$ of the subsurface over the period of one year. While the thawed ground fraction $D_{\mathrm{thaw}}^{3\mathrm{m}} \in [0,1]$ is harder to retrieve from field observations than thaw depths, it is more informative
as a first-order approximation of conditions favourable for microbial decomposition of organic matter. For Fig. 8, we derived a simple approximation of this quantity, based on the Stefan equation (see Appendix B).

- Similar to the deep thermal contraction cracking, we diagnosed the *shallow thermal contraction cracking activity* ($a_{\mathrm{FC}}^{\mathrm{shallow}}$), but used different thresholds of $T_{1\mathrm{m}} < -5\,°\mathrm{C}$ and $\nabla T_{1\mathrm{m}} > 7\,°\mathrm{C}\,\mathrm{m}^{-1}$ (i.e., $T_{0\mathrm{m}} < T_{1\mathrm{m}} - 7\,°\mathrm{C}$). Shallow thermal contraction cracking can also occur outside permafrost-affected areas.

The corresponding results for these diagnostics are provided in Figures A1, A2, and A3 below.

## Appendix B: Calculation of annual thawed ground fraction using the Stefan model

For Fig. 8 we calculated the annual thawed ground fraction ($D_{\mathrm{thaw}}^{d}$) within the upper $d = 3\,\mathrm{m}$ of the subsurface in dependence of the mean ($\bar{T}_{\mathrm{a}}$) and the seasonal amplitude ($\tilde{T}_{\mathrm{a}}$) of the surface temperature using the Stefan model (Riseborough et al., 2008). For this, we assumed the annual temperature evolution to have a sinusoidal shape:

$$T_{\mathrm{a}}(t) = \bar{T}_{\mathrm{a}} + \frac{\tilde{T}_{\mathrm{a}}}{2} \cdot \sin\left(2\pi \frac{t}{P}\right) \tag{B1}$$

where $t$ is the time and $P = 365\,\mathrm{d}$ is the period length (Fig. B1 a).

The thawing (freezing) index $I_{\mathrm{tlf}}(t)$ was determined as the integral over the positive (negative) surface temperature, scaled using a thawing (freezing) -season $n$-factor ($n_{\mathrm{tlf}}$). The integration limits were set to start at the first root of the temperature





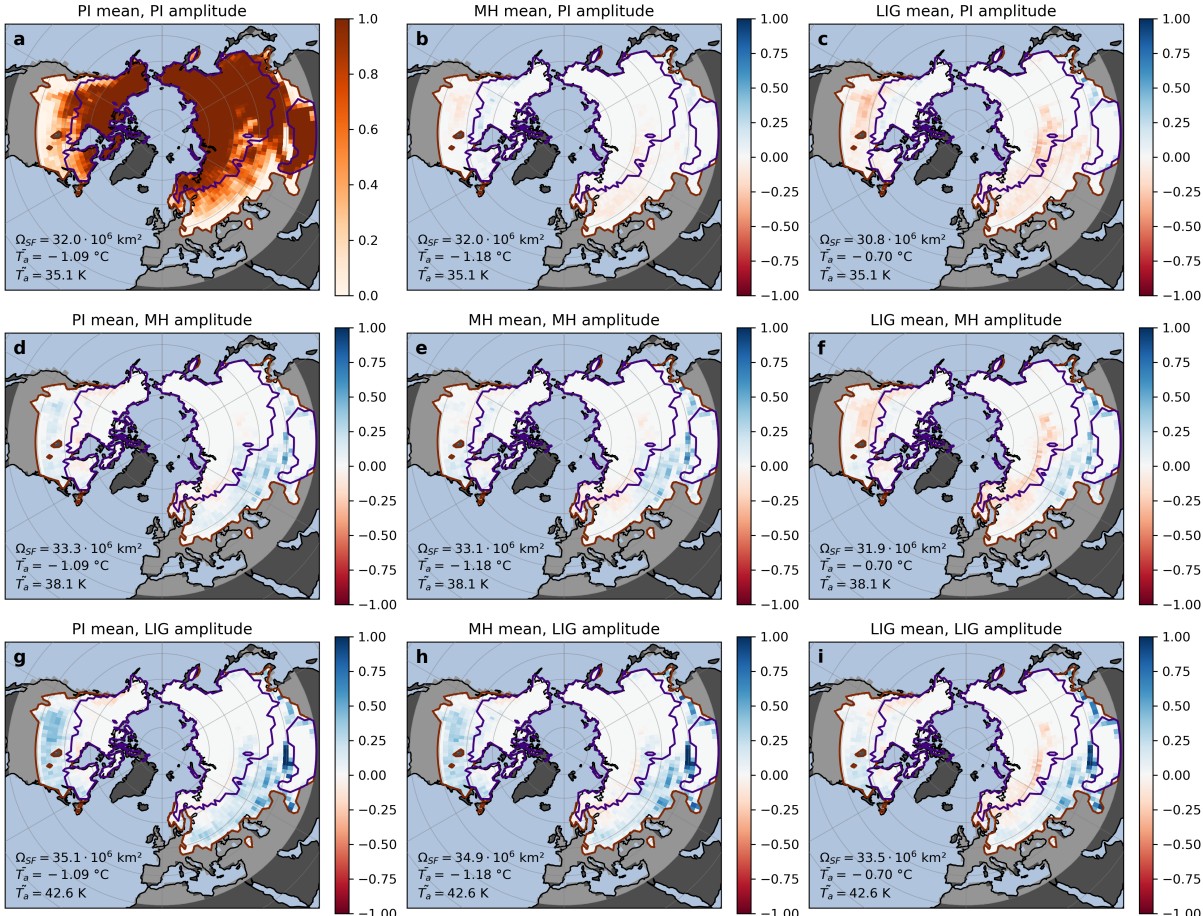

**Figure A1.** Simulated seasonal frost probabilities ($p_{SF}$) for all climate forcings, shown as absolute values for the PI experiment in panel a, and as anomalies for all other experiments. The results for the original climate forcings are shown on the on the diagonal (PI: a; MH: e; LIG: i), and the off-diagonal panels show results for the alternative climate forcings, with each column having the same mean temperature (PI: a,d,g; MH: b,e,h; LIG: c,f,i) and each row having the same temperature amplitudes (PI: a,b,c; MH: d,e,f; LIG: g,h,i). The purple (orange) line marks the limits of the joint permafrost (seasonal frost) domain of all simulations. The seasonal frost extent ($\Omega_{SF}$), the mean annual temperature ($\bar{T}_a$), and the mean temperature amplitude ($\tilde{T}_a$) for the joint seasonal frost domain are given in the lower left corner of each panel.

curve at $t_0 = -\frac{2\pi}{P}\arcsin\left(\frac{2\bar{T}_a}{\tilde{T}_a}\right)$ (Fig. B1 a,b):

$$I_t(t) = n_t \int_{t_0}^{t} T_a(t')|_{T_a>0}\, dt' \tag{B2}$$

$$I_f(t) = -n_f \int_{t_0}^{t} T_a(t')|_{T_a<0}\, dt' \tag{B3}$$

$$\tag{B4}$$



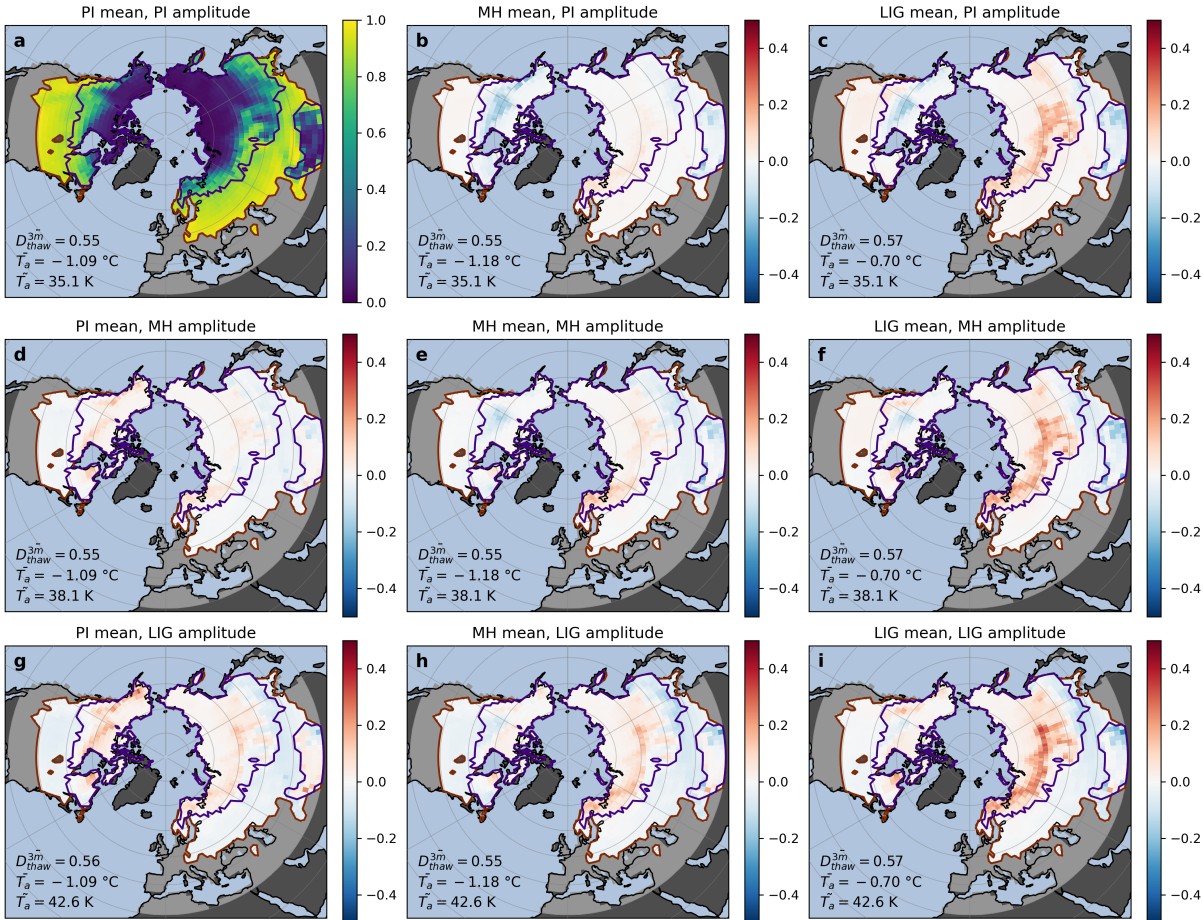

**Figure A2.** Same as Fig. A1, but for the annual thawed ground fraction ($D_{\text{thaw}}^{3\text{m}}$). The mean annual thawed ground fraction ($\bar{D}_{\text{thaw}}^{3\text{m}}$), the mean annual temperature ($\bar{T}_a$), and the mean temperature amplitude ($\tilde{T}_a$) for the joint seasonal frost domain are given in the lower left corner of each panel.

With these, the progression of the thaw (freeze) front $X_{\text{tlf}}(t)$ was determined using the Stefan equation (Fig. B1 c):

$$X_{\text{tlf}}(t) = \sqrt{\frac{2\,k_{\text{tlf}}\,I_{\text{tlf}}(t)}{\theta\,\phi\,L_{\text{sl}}}} \tag{B5}$$

where $L_{\text{sl}} = 334 \times 10^6 \,\text{J m}^{-3}$ is the volumetric latent heat of fusion of water, $\theta$ the water/ice saturation of the sediment's pore space, $\phi$ the sediment's porosity. $k_{\text{tlf}}$ refers to the thawed/frozen ground thermal conductivity parameterized as follows:

$$k_{\text{tlf}} = \left(\theta\phi\sqrt{k_{\text{wli}}} + (1-\phi)\sqrt{k_s}\right)^2 \tag{B6}$$

with the constituents' thermal conductivities set as $k_s = 3.0 \,\text{W K}^{-1}\,\text{m}^{-1}$ for the sediment, $k_i = 2.2 \,\text{W K}^{-1}\,\text{m}^{-1}$ for ice, and $k_w = 0.57 \,\text{W K}^{-1}\,\text{m}^{-1}$ for water.



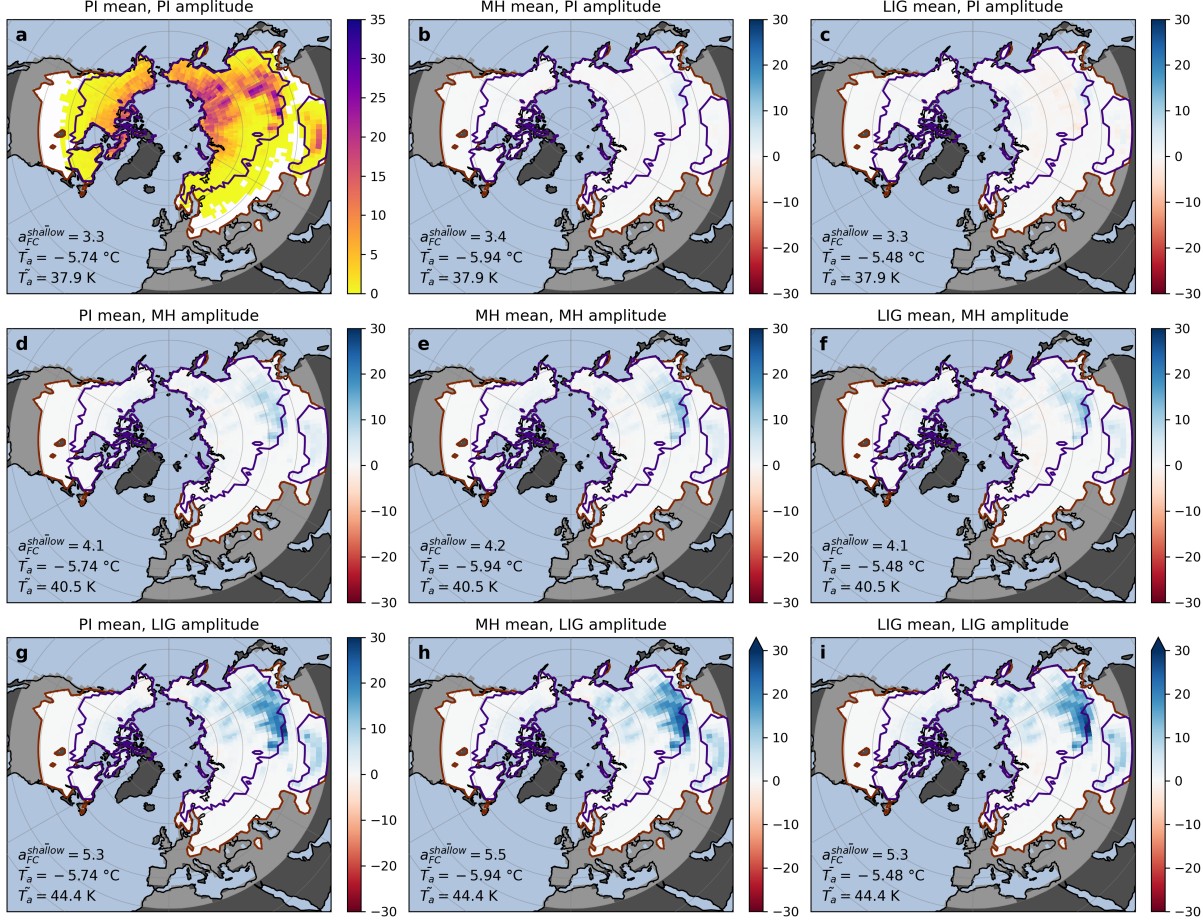

**Figure A3.** Same as Fig. A1, but for the shallow thermal contraction cracking activity ($a_{\mathrm{FC}}^{\mathrm{shallow}}$). The mean shallow thermal contraction cracking activity ($\bar{a_{\mathrm{FC}}^{\mathrm{shallow}}}$), the mean annual temperature ($\bar{T}_{\mathrm{a}}$), and the mean temperature amplitude ($\tilde{T}_{\mathrm{a}}$) for the joint seasonal frost domain are given in the lower left corner of each panel.

The Stefan model indicates thermally stable permafrost, if the maximum freeze depth exceeds the maximum thaw depth over the course of one year ($X_{\mathrm{f}}(t_0 + P) > X_{\mathrm{f}}(t_0 + P)$); otherwise there is no permafrost. For the permafrost case, the annual thawed ground fraction $D_{\mathrm{thaw}}^{d}$ was obtained by integrating the difference between the thaw and freeze fronts between $t = t_0$ and the point $t = t_1$ where the freeze front reaches the thaw front, and normalizing by the maximum depth $d$ and temporal period $P$ (Fig. B1 c):

$$D_{\mathrm{thaw}}^{d} = \frac{\int_{t_0}^{t_1} \min[d,\, X_t(t') - X_f(t')]\,\mathrm{d}t'}{d\,P} \qquad \text{if} \qquad X_{\mathrm{f}}(t_0 + P) > X_{\mathrm{f}}(t_0 + P) \qquad \text{(permafrost)} \qquad \text{(B7)}$$

For the case of no permafrost, the freeze and thaw fronts were shifted such that the freezing starts at $t = t_0$. Then, $D_{\mathrm{thaw}}^{d}$ was obtained by first determining the annual frozen ground fraction through a similar integration as above, and then subtracting it



from 1:

$$D_{\text{thaw}}^d = 1 - \frac{\int_{t_0}^{t_1} \min[d, X_f(t') - X_t(t')] \, dt'}{d\,P} \qquad \text{if} \qquad X_{\text{f}}(t_0 + P) \leq X_{\text{f}}(t_0 + P) \qquad \text{(no permafrost)} \tag{B8}$$

Note that $D_{\text{thaw}}^d = 0$ if $\bar{T}_{\text{a}} + \frac{\tilde{T}_{\text{a}}}{2} < 0$, and $D_{\text{thaw}}^d = 1$ if $\bar{T}_{\text{a}} - \frac{\tilde{T}_{\text{a}}}{2} > 0$. For Fig. 8 we set $n_{\text{t}} = 1.0$, $n_{\text{f}} = 0.5$, $\phi = 0.5$, and $\theta = 0.75$, such that $k_{\text{t}} \approx 1.32\,\text{W}\,\text{K}^{-1}\,\text{m}^{-1}$ and $k_{\text{f}} \approx 2.02\,\text{W}\,\text{K}^{-1}\,\text{m}^{-1}$.

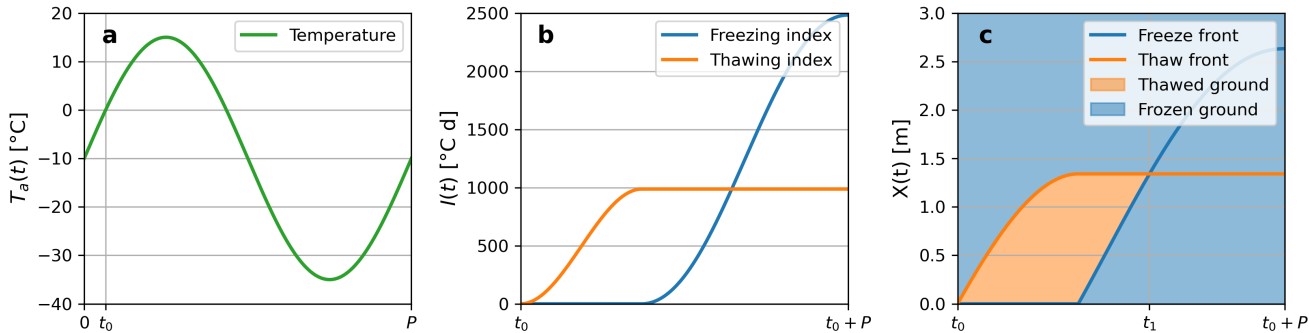

**Figure B1.** Illustration of how the thawed ground fraction ($D_{\text{thaw}}^d$ was obtained using the Stefan model. a: We assumed a sinusoidal temperature curve $T_{\text{a}}(t)$ of mean $\bar{T}_{\text{a}}$ and amplitude $\tilde{T}_{\text{a}}$. b: Thawing (freezing) indices ($I_{\text{tlf}}(t)$ were obtained by integrating the positive (negative) temperatures and scaling them using $n$-factors. c: The thawed ground fraction was obtained by integrating the difference between the thaw and freeze fronts ($X_{\text{tlf}}$), and normalizing it by the depth $d$ and period of one year ($P$).

*Author contributions.* J. N. conceived the study, performed the CryoGridLite simulations, analyzed the results, and led the manuscript preparation. M. L. advised on the methodology secured funding. L. A. M.-I. compiled the speleothem and pollen data and conducted the model evaluation. E. D. advised on the paleo records for model evaluation. M. W. advised on the methodology and secured funding. All
authors discussed the results and edited the manuscript.

*Competing interests.* The authors declare no competing interests.

*Acknowledgements.* We thank Yuchen Sun for providing the AWI-ESM-2.5 climate forcing data for the PI, MH, and LIG. We thank Christian Stepanek for providing AWI-ESM-2.1 output for the historical period and the SSP scenarios. We thank Wenjia Li for providing the pollen data. We thank Paul Gierz for assistance with the AWI-ESM-2/CryoGridLite setup.





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
