# Peer review of "How temperature seasonality drives interglacial permafrost dynamics: Implications for paleo reconstructions and future thaw trajectories"

_EGUsphere, 2024_

## Author Comment (AC1)

Reviewer comments are in *black italics,* our responses are in blue font.

**Reviewer 1**

*This study highlights the importance of seasonality in permafrost dynamics. I think that this is a good paper, including clear logic and professional writing. The results will be important for permafrost research. My comments are the following.*

We thank the reviewer for the overall positive assessment of our study and the numerous suggestions for improvements. Below, we provide point-by-point replies to the comments made by the reviewer.

*L12, some readers may be interested in how annual mean and seasonal amplitudes changes, here.*

We extended the abstract according to the reviewer's suggestion and provide numbers.

*L104, what is the deepest soil column of simulation for CryoGridLite permafrost model?*

The lower boundary of the model is at a depth of 500m where a constant geothermal heat flux is applied. We specified this in the revised version.

*L37, "Brierley et al. (2020); Otto-Bliesner et al. (2021)" should be (Brierley et al., 2020; Otto-Bliesner et al., 2021).*

Thanks. We fixed this typo in the revised version.

*L170, the authors say that speleothem growth suggests absence of permafrost here, but in L180 the authors say that locations with speleothem growth is agreement with the model if permafrost probability <90%. The former and the latter appear to be contradictory.*

Indeed, the formulation in the manuscript reflects our approach correctly. For simulated permafrost probabilities between 10% and 90% (corresponding to the zones of sporadic and discontinuous permafrost), we considered reconstructions of both speleothem growth and hiatuses to be in plausible agreement with the simulations. Please note that this concerns only a relatively small portion of the simulation domain, while the majority of the domain was simulated to be either permafrost-free or continuous permafrost (Fig. 3).

*Figure 4, remove a "on the", it is repeated.*

   Thanks. We fixed the typo.

*Figure 5, I suggest to add the significance of these correlations.*

   We appreciate this suggestion and have marked statistically significant correlations (p<0.05) in the figure by highlighting the correlation value in bold. We have also revised the respective results descriptions to mention the statistical significance of the correlations.

*Lines 240-245, the phenomenon is clear, i.e., mean temperature control the permafrost area rather than seasonal temperature amplitude. However, what are the physical explanations?*

   In this paragraph, we were mainly describing the simulation results regarding how permafrost extent is correlated with mean temperatures and seasonal amplitudes, respectively. We discuss these results in the first paragraph of section 4.2.1.  where we also put it into the context of earlier findings. In the revised manuscript, we have slightly extended this discussion, to also shed light on the physical explanation.

*L320 and L324, what is the difference between "a mean global surface warming above recent conditions" and "MH-PI global temperature anomaly"?*

   The former refers to the temperature difference of +1°C between the MH and "modern values" which was assumed by Anisimov and Nelson (1996). The latter refers to the differences between the MH and the pre-industrial reference period for which we found several estimates in the literature, in particular those references in Kaufman et al. (2023). We think that the value of Anisimov and Nelson (1996) can be compared to the MH-PI anomalies provided in Kaufman et al. (2023) despite slightly different definitions.

*L467-471, here, I suggest to state that whilst the most recent interglacial climates providing less analogues for the future with respective to permafrost dynamics, this do not exclude that the past older warming period may be appropriate. For instance, the most simulation study (https://doi.org/10.1073/pnas.2301954120) on the mid- Pliocene warm period (mPWP, ~3.264 to 3.025 Ma) permafrost. During the mPWP, the temperature increase significantly in both winter and summer. The period also evolves regional differences in warming, in particular in the high latitudes. These are similar to the future warming. So, mPWP permafrost remains one of the analogs for the future permafrost dynamics, and likely the results (highly restricted extent) has implications for the future.*

   We appreciate this suggestion and will extend the discussion accordingly.

---

## Author Response (AR1)

8 July 2025

Dear Alberto Reyes,

Thank you for handling our manuscript and the invitation to resubmit it after minor revisions. We have carefully taken all comments by the two reviewers into consideration and incorporated the changes as outlined in the replies. We have also taken up your suggestion and slightly extended the discussion of evidence for past thaw vs persistence. We hope that this aligns with what you had in mind.

Please see the track-change version as well as the previous replies for the point-by-point responses and all changes made to the manuscript.

We look forward to your decision and the potential publication of our article in CP.

Kind regards,

Jan Nitzbon (on behalf of all authors)